# Involving Children in Creating a Healthy Environment in Low Socioeconomic Position (SEP) Neighborhoods in The Netherlands: A Participatory Action Research (PAR) Project

**DOI:** 10.3390/ijerph182212131

**Published:** 2021-11-19

**Authors:** Lisa Wilderink, Ingrid Bakker, Albertine J. Schuit, Jacob C. Seidell, Carry M. Renders

**Affiliations:** 1Department of Health Sciences, Faculty of Sciences, Amsterdam Public Health Research Institute, Vrije Universiteit Amsterdam, 1081 HV Amsterdam, The Netherlands; j.c.seidell@vu.nl (J.C.S.); carry.renders@vu.nl (C.M.R.); 2Department of Healthy Society, Windesheim University of Applied Sciences, 8017 CA Zwolle, The Netherlands; i.bakker@windesheim.nl; 3School of Social and Behavioral Sciences, Tilburg University, 5037 AB Tilburg, The Netherlands; jantine.schuit@tilburguniversity.edu

**Keywords:** socioeconomic health inequalities, participatory action research, health behavior, children

## Abstract

To ensure that health behavior interventions for children living in low socioeconomic position (SEP) neighborhoods are in line with children’s wishes and needs, participation of the children in the development, implementation, and evaluation is crucial. In this paper, we show how children living in three low-SEP neighborhoods in the Netherlands can be involved in Participatory Action Research (PAR) by using the photovoice method, and what influences this research process. Observations, informal chats, semi-structured interviews, and focus group discussions with children and professionals were done to evaluate the research process. The photovoice method provided comprehensive information from the children’s perspectives. With the help of the community workers, the children identified feasible actions. We found that it is important to constantly discuss the research process with participants, start with a concrete question or problem, and adapt the project to the local context and skills of participants.

## 1. Introduction

Reducing socioeconomic health inequalities is a challenge for health policymakers and civil society. All over the world, people with a relatively low socioeconomic position (SEP) in their society have a shorter life in good health compared to people with a high SEP [1,2]. This is partially linked to the fact that the physical and social environment of people with a relatively low SEP offers fewer opportunities and less support for a healthy lifestyle. Low SEP neighborhoods are usually characterized by, for example, less public recreation space to be physically active, more marketing of unhealthy products, and a higher density of fast-food restaurants [3,4]. Moreover, people with a relatively low SEP often experience less social support (e.g., prevailing norms and example behavior) from peers and family for healthy behavior [5]. From an early age onwards, children living in low-SEP neighborhoods in the Netherlands and other Western countries have less opportunities to be physically active and/or to consume a healthy diet. They generally show higher rates of overweight and obesity than children from higher SEP groups [6].

The influence of the neighborhood environment on (un)healthy behavior has been widely described in previous literature [7,8,9,10,11]. The concept of the obesogenic environment emphasizes the effect of environmental factors that support unhealthy behavior [12,13,14]. Changes in this environment, however, can contribute to increased physical activity and healthy dietary behavior [15]. For example, making the environment more attractive for physical activity or providing additional sporting facilities contributes to more physical activity [15,16]. Moreover, the availability of playground equipment [17] and public green space [18] contributes positively to healthy behavior. Changing the environment in low SEP neighborhoods is, therefore, a way to stimulate healthy behavior and possibly reduce socioeconomic health inequalities.

In the context of socioeconomic health inequalities, it is important to distinguish between the terms equality and equity. Equality means giving or treating everyone the same, whereas equity means giving people what they need to reach their best health, based on the needs of the recipients [19]. According to the World Health Organization (WHO), equity is ‘the absence of avoidable or remediable differences among groups of people, whether those groups are defined socially, economically, demographically or geographically.’ [20]. Achieving health equity requires more than treating everyone equally: it requires treating everyone according to their circumstances and needs. Most health education approaches and interventions have a smaller impact in populations with a low SEP than in more privileged populations [21]. An important reason for this is that people with a low SEP are not sufficiently reached because interventions often do not properly match their interests, needs, and experiences [22,23]. In spite of good intentions, such interventions maintain or even increase socioeconomic health inequalities instead of reducing them.

Participatory Action Research (PAR) is a citizen science approach well suited to contribute to the investigation of complex public health problems such as socioeconomic health inequalities, because, through participation, the research process is led by the circumstances, wishes, and needs of the people involved [24]. It acknowledges that the target group has the unique expertise required to better understand and address a complex health problem and takes different perspectives on both causes and solutions of the health problem into account. The multiple perspectives of people representing the target group support the development and implementation of interventions tailored to the needs, wishes, opportunities, literacy, and skills of the people involved. This tailored development and implementation can in combination increase the impact on reducing socioeconomic health inequalities [25,26]. PAR has been shown a feasible and promising approach to study the neighborhood environment in low SEP communities and find leverage points for change to promote healthy behavior [27,28].

Nickel and Von dem Knesebeck [29] show in their systematic review of community-based health promotion interventions that participation is a key component for the effectiveness and sustainability of community-based approaches and interventions. Participation of people from low-SEP groups specifically, therefore, contributes to the design of approaches and interventions that are effective for them. Moreover, participation of the target group in all phases of the research process can create a sense of ownership and empowerment by improving participants’ capability to be advocates for changes in their own circumstances [30,31].

PAR has been gaining status in public health science since the 1990s [25]. In addition, Jacquez and Vaughn [32] showed in their review of children and adolescents in community-based participatory research that there is also a growing interest in the participation of children in research. To ensure that health behavior interventions for children living in low-SEP neighborhoods are in line with children’s wishes, needs, talents, motivations, and opportunities in their environment, participation of the children in the development, implementation, and evaluation of those interventions is crucial [33]. Previous studies show the added value of involving children as researchers, to understand their perspectives better [31,34], understand their vision on environmental influences more in depth [35], and the benefits that children gain from participating themselves by increasing their knowledge and skills to be advocates for change [36]. As mentioned by Mitchell, Kearns [37], children are experts on their own lives and for that reason should be empowered to play a prominent role in research about their lives.

The amount of literature on the participation of children in research, in general, is growing, however, research which explicitly reflects critically on the process of including children in all stages of the research process is still limited [38,39,40].

This study is a qualitative evaluation of a local PAR project (‘project MAPZ’, Dutch acronym for ‘Thinking, Active and Participating Zwolle’), using the photovoice method, on encouraging healthy dietary and physical activity behavior in children in three low-SEP neighborhoods in Zwolle, the Netherlands. Project MAPZ is part of the Zwolle Healthy City approach, an integrated community-based approach aimed at reducing socioeconomic health inequalities. Since 2010, several local policy and community-based organizations in Zwolle have committed themselves to the approach in order to encourage healthy dietary and physical activity behavior in children. However, the participation of children in the design, implementation, and evaluation of interventions and activities within this approach has lagged behind [33]. Improving children’s participation in design, implementation, and evaluation of health behavior interventions could lead to better alignment of the activities within the approach with the children’s wishes, needs, talents, motivations, and opportunities. This could result in a larger impact of those activities (i.e., more healthy dietary and physical activity behavior), and may eventually contribute to the reduction in socioeconomic health inequalities.

In this paper, we show how children living in low-SEP neighborhoods can be involved throughout the research process, using the photovoice method. We describe the methodological steps undertaken in the participatory process and what influenced this process according to the children, community workers, and researchers. We aim to share what we have learned and to inspire other researchers and communities that want to involve children in creating a healthy neighborhood environment and encouraging healthy behavior.

## 2. Methods and Procedures

### 2.1. Setting

Project MAPZ (Dutch acronym for ‘Thinking, Active and Participating Zwolle’) is a Participatory Action Research (PAR) project aimed at creating a healthy neighborhood environment, and with that, encouraging healthy dietary and physical activity behavior in children living in low-SEP neighborhoods. Researchers, children, and professionals working for a local welfare organization collaborated on the project. The aim of project MAPZ was to improve the participation of children aged 8 to 12 living in low-SEP neighborhoods by using the photovoice method and involving them from intervention design to implementation and evaluation.

The three neighborhoods in which project MAPZ was initiated are located in Zwolle, a medium-sized municipality (around 125,000 citizens) in a rural area in the northeast of the Netherlands. The three neighborhoods have a relatively large percentage of low-income households (an income based on purchasing power and lower than the social assistance benefit): 73.4% in neighborhood A, 62.4% in neighborhood B, and 60.4% in neighborhood C, compared to 41.8% low-income households in the entire municipality of Zwolle. Neighborhood C is a more multicultural neighborhood (28.5% non-western migrants) compared to neighborhood A (17.7%) and neighborhood B (15.2%) [41]. The percentage of overweight children is 23% in neighborhood C and 25% in both neighborhoods A and B, compared to 14% in the total municipality of Zwolle [42].

In those low-SEP neighborhoods, the local welfare organization continually organizes activities in community centers which every child in the neighborhood can join for free. The community workers focus on offering leisure activities that contribute positively to the youths’ lives, e.g., through skills development, help with personal or social problems, and promoting active participation in society. Those weekly meetings focused on project MAPZ for several months in 2019, ranging from three months in neighborhood B, eight months in neighborhood A, to ten months in neighborhood C, depending on how it fitted within the organization. The community workers facilitated the PAR meetings together with the researchers.

The study protocol was approved by the Medical Ethics Review Committee (METC) of VU University Medical Center and confirmed the study did not require medical ethical approval under Dutch legislation on medical trials (2018.601).

### 2.2. Participants

The three participating groups were pre-existing groups of children that voluntarily come together after school in the community centers. A total of 28 children were involved throughout the process of retrieving information and performing actions that fit their perceptions of a healthy neighborhood environment. The participating group in neighborhood A and the participating group in neighborhood C were mixed in terms of gender, while the participating group in neighborhood B consisted of girls only (see Table 1). The participating group in neighborhood C consisted only of children with a non-western migrant background. The PAR meetings were facilitated by a total of two (female) researchers and five community workers working for the local welfare organization (four female, one male), spread across the three neighborhoods. One principal researcher (LW) was always present in all three neighborhoods while the supporting researcher only participated in neighborhoods A and B. In every neighborhood, one or two community workers participated.

In neighborhoods B and C, all children who visited the after-school activities participated in project MAPZ (*n* = 11, *n* = 7). The setting was school-like: in an enclosed space. In neighborhood A, the after-school activities were in a freer setting: in the community center where people could walk in and out, and children could also choose to do other activities. Children could choose to participate in project MAPZ (*n* = 10).

### 2.3. Procedures

Photovoice, a participatory action research methodology, was used to identify barriers and facilitators for healthy dietary and physical activity behavior. In previous research, photovoice has been shown a feasible method to find leverage points in the neighborhood environment to stimulate the healthy behavior of children [43,44]. Because children spend a lot of their time in the neighborhood, it is useful to understand how this environment stimulates their healthy behavior. According to Bashore et al. (2017), photovoice is a powerful tool that can be used with the most vulnerable youth to allow them to share their view of their environment [35]. Photovoice is known as an eligible method to engage children who are in general less literate and prefer taking action over talking or listening [36,45]. The method involved children taking photos, having discussions about them during focus group meetings, and eventually inspiring the participating children to bring about change in the neighborhood that addressed the barriers or facilitators [46]. Taking photos and showing them to the group helped the children to create their own narrative, without bias from the researchers [34]. As in previous research, a simplified content analysis approach with the aim to derive themes was used to analyze the photos and discussion transcripts [35]. Photovoice made it possible to record and represent children’s everyday realities. In this way, children could function as key informants in their neighborhoods [47]. In our study, the photovoice method enabled children living in low-SEP neighborhoods to give a more in-depth insight into their daily lives and the social and physical environment that influences their healthy behavior.

#### 2.3.1. Step 1—Getting Started

In order to involve children living in the low-SEP neighborhoods, collaboration was initially sought with the welfare organization that organizes after-school activities for children. The photovoice protocol was developed with the participating community workers in order to let the project fit the neighborhood-specific context, e.g., how often and where meetings would take place with the children. Prior to the start of project MAPZ, the researchers participated in weekly activities to get to know the children and build trust with them. To enthuse the children for the project, they were involved from the beginning: a name and logo for the project were developed in co-creation. Project MAPZ was introduced to the children as a project in which children themselves could make their neighborhood more fun, beautiful and healthy. A website and information flyers for children and parents were designed. As low literacy is common in low-SEP communities [48], all written information was checked for readability for low-literate people by a panel of formerly low-literate people and adapted accordingly.

Written informed consent was obtained from the children and their parents beforehand. The informed consent was about agreement on participating in the project, knowing what the project is about, knowing that the children can choose whether they want to participate or not, and knowing that they can always stop participating. Children and parents also gave permission for using the (photo) data gathered in the project and recording the discussions. The participating children signed the informed consent in the first meeting. The informed consent for the parents together with an information flyer was given to the children to bring home in the first meeting and received signed back in the second or third meeting.

#### 2.3.2. Step 2—Taking Photos

In step 2, the researchers informed the children about the procedure and objectives of project MAPZ. The children and researchers had a discussion on the ethics of photovoice, which was framed for the children as agreeing on behavioral ‘rules’ or principles. Taking into account the privacy of others, the children agreed to take photos only in public areas and not to photograph identifiable people. The children received a notepad with instructions in text and images and a name badge which said ‘junior researcher’. The children walked around the neighborhood in small groups of three or four and took photos with a loaned photo device. The assignment was to (1) ‘take photos showing where you like to play, exercise, eat, drink and relax in your neighborhood’, (2) ‘take photos of things you like and dislike in your neighborhood’ and (3) ‘take photos of things in your neighborhood that make you happy and things you are worried about’. The children wrote down a short description of the photos they took in their notepad, including why and where the photo was taken, to help them remember this when the photos were discussed a week later.

#### 2.3.3. Step 3—Identifying Themes

One week later, a focus group discussion with the same groups of children (*n* = 7–11) was held and audio-recorded. One child in every group was assigned as ‘Chief of Recordings’ and responsible for the recording of the focus group discussions to increase their feeling of being a member of the research team. Photos were printed, which made it easier for the children to discuss them while showing them to the researchers and other children. Children showed their best photos or photos that were, according to them, the most important to discuss. During the discussion of each photo, the children were asked multiple questions by the researcher(s) in order to help them clarify the social or physical barrier(s) or facilitator(s) for healthy dietary and physical activity behavior in their living environment (see Table 2). The community workers and other participating children were given the opportunity to ask questions as well.

The children participated in the analysis by finding patterns and relationships in the photo data set. Together with the researchers, the children performed a simplified version of thematic content analysis by clustering the photos to derive underlying themes. In neighborhood A, four themes were derived after discussion, in neighborhood B two and in neighborhood C four.

#### 2.3.4. Step 4—Creating Posters to Draw Conclusions

In the fourth step, children created posters for the ten themes containing the relevant photos, drawings, stickers, and written explanations from the children. Subsequently, the children, researchers, and community workers had a focus group discussion about the importance of the different themes and what could be concluded from the posters.

#### 2.3.5. Step 5—Identifying Feasible Actions

The children, researchers, and community workers brainstormed together in step 5 on how to translate the conclusions into actions that could contribute to change in the neighborhood environment. The children were asked to come up with several actions per conclusion that could be carried out, either by themselves or by others, such as the local government. The children voted for the most feasible actions for implementation.

### 2.4. Data Collection and Analysis

Through the photovoice method, barriers and facilitators for healthy dietary and physical activity behavior were identified. Information on how the project MAPZ proceeded was iteratively collected from researchers’ observations and informal chats after the meetings with the children and community workers, which made reflection and adjustment during the process possible. This was all recorded in a logbook by the principal researcher (LW). Semi-structured interviews with the community workers and supporting researcher and focus group discussions with the children at the end of the project, were held to evaluate the complete PAR project (see Table 3).

#### 2.4.1. Focus Group Discussions

Focus group discussions with children were held after the photovoice steps to evaluate project MAPZ in neighborhoods A and C. This could not take place in neighborhood B due to the resignation of the community worker who facilitated the meetings in neighborhood B. Children were asked about their experiences with the project. They were asked to write down or draw what they liked and disliked on a sticky note for every step of the photovoice process. Next, they were asked to elaborate on what they wrote or draw on the sticky notes. The focus group discussions were audio-recorded and transcribed verbatim anonymously. The focus groups were conducted by a researcher who was experienced in conducting focus group discussions (LW). Another researcher observed the focus group and took notes about the verbal and non-verbal communication between the researcher and the children and the ambiance in the group (see below). The duration of the focus group discussions ranged from 35 to 52 min. The focus group discussions were transcribed verbatim and analyzed by LW to understand the experience of the children and aspects that influenced the photovoice process, according to them.

#### 2.4.2. Semi-Structured Interviews

The principal researcher (LW) interviewed five community workers working for the welfare organization and the supporting researcher at the end of the process to evaluate project MAPZ. In a semi-structured interview, they were asked about their experiences with the project. The interview guide consisted of chronologically arranged questions for every specific step as ‘What went well?’, ‘What could be done better next time?’ and ‘What is needed for that?’ related to themes such as planning, communication, participation, and collaboration. The duration of the interviews ranged from 49 to 92 min. Written informed consent was obtained from the respondents beforehand. The semi-structured interviews were transcribed verbatim and analyzed by LW to distract the experience of the community workers and involved researcher, and aspects that influenced the PAR process according to them.

#### 2.4.3. Observations and Informal Chats

Observations that were made during the photovoice meetings were recorded in a notebook. Observations were made about the general atmosphere, attitudes towards the aim of the project, and attitudes towards the (process of the) specific meeting. During informal chats at the end of every meeting, the children and community workers were asked questions such as ‘What went well?’ and ‘What could be improved next time?’. Data analysis of those observations and informal chats started while data collection was still in progress, which made reflection and adjustment of the process possible. The notes that resulted from the observations and informal chats were eventually included in the analysis of the focus group discussions and semi-structured interviews to come to the following results.

## 3. Results

The results include a presentation of the lessons learned in the focus group discussions, semi-structured interviews, observations, and informal chats evaluating project MAPZ according to the steps of the project.

### 3.1. Step 1—Getting Started

The supporting researcher who was interviewed indicated that collaborating with children who visited the ongoing after-school activities was appropriate and worked well. According to her, it was a desirable way to recruit this low-SEP target group that does not usually find it easy to talk about their wishes, needs, motivations, and talents for healthy dietary and physical activity behavior. The community workers also indicated that collaborating with the welfare organization worked well. They stated that as community workers, they were familiar with the children and would know their preferences for activities since they worked with them on a daily basis. According to them, this led to a feasible project, adjusted to the local context. The community workers indicated that it was useful to discuss the protocol with them before the start of the project: ‘*The conversation we had with you* [the principal researcher] *in advance was very nice, we made adjustments together that are beneficial to the activities and the target group*.’ (Community worker). For example, the idea of using Polaroid cameras to take photos instead of using digital cameras was a suggestion from a community worker in this conversation before the start of the meetings with the children. This was suggested because Polaroid cameras were popular among children and using them would make these children more enthusiastic about the project. This was an alteration that was carried out while the project had already started and influenced the process. Another positive aspect of using the Polaroid cameras, according to the researchers, was that the children had to think critically about what they were going to photograph since it is not possible to take an infinite number of photos with this non-digital camera.

In the researchers’ experience, it was good to be involved with the children and see the children on a regular basis before starting the project. In project MAPZ, this led to familiarity with the researchers among the children: *‘I think that the thing that worked out the best for us was that we were involved with the group quite early on and that the children really knew us and that they really wanted to tell more.’* (Supporting researcher). It also worked well in enthusing the children to give them the feeling that they were researchers, for example, by giving them ‘junior researcher’ badges and emphasizing that they were needed in the project. The children were told that the researchers did not know what it is like to be a child living in that specific neighborhood and that through participation in the project, the children could provide insight into their lives.

### 3.2. Step 2—Taking Photos

The participating children indicated that it was fun to walk through the neighborhood in small groups to take the photos: *‘I like taking pictures!’* (Child). According to the children, participating in the project was fun; it was different from school and being at home. The community workers also confirmed that taking photos with the Polaroid camera was a fun way for children to participate: ‘*It just has to be something that they don’t do every day. Those type of activities are often popular.’* (Community worker). The community workers indicated that the children did not want to listen and talk too much but would prefer to take action: ‘*Where they have to use their hands and feet and get moving. Get out of their chairs and school desks, away from that formal meeting structure.’* (Community worker). For that reason, going outside in groups to take photos worked well.

The community workers indicated that photovoice was an expedient method but that a specific question or problem is needed. Starting by saying ‘take photos of places you like to visit in the neighborhood’ was too vague for children, according to the respondents. They did not understand the underlying question or the problem: *‘I think what has made it difficult, normally we see something, something is going on, then you’re going to do something about it. And now it was actually… we have a group, there was a theme, and you investigate whether something is really going on. That is actually the other way around*.’ (Community worker). The question or problem being researched must be specifically formulated and not too broad, according to the respondents. A community worker indicated that the ‘unhealthy lifestyle’ theme was too broad. On her advice, we specified that to the themes of ‘being active outside’ and ‘eating/drinking’.

### 3.3. Steps 3 and 4—Identifying Themes and Creating Posters to Draw Conclusions

The community workers indicated that the focus group discussion meetings in steps 3 and 4 of 1.5 h were long, and to keep the children engaged, the meetings should not contain only serious activities. Alternating the conversations on identifying themes and drawing conclusions with games or using creative methods such as drawing or creating posters worked well in this regard, according to the researchers.

According to the children, the printed photos helped to talk about things in their neighborhood: *‘Then I can say: look, that’s there, or there.’* (Child). Moreover, in the experience of the other respondents, being able to show the printed photos helped because it was easier to talk and remember. The notes in the note blocs that the children made helped as well in this regard.

It took more time and effort to let the youngest children (those who were eight or nine years old) participate in a feasible way in the project, compared to the older children, according to the researchers and community workers. It was more difficult for children of that age to understand what they were asked to do: ‘*If you look at the development of children, you actually see that from the age of ten or so they start thinking about their environment or the children around them.’* (Community worker). As a result, it took more time to explain the assignments within the project to those young children and help them express their vision on improvements.

In step four, the researchers concluded that not much information was gained during the focus group discussion about the photos in neighborhood C in comparison to neighborhoods A and B. According to the community workers, this was partly due to the skills of participating children. The group consisted of children with a migrant background and refugee children, who found it harder to express their perceptions in words. Because less information was derived in these focus group discussions, a survey was conducted among their classmates to verify the derived themes and gather more information on their view on their (un)healthy neighborhood. This was suggested by the participating children. The children formulated the survey questions they wanted to ask. The survey consisted of questions on, for example, what sports classmates would like to join in the neighborhood and their opinion on specific playgrounds. The participating children conducted the surveys among their classmates (4 classes, 112 children in total) and analyzed the results together with the researchers. Children participated in the analysis of the surveys by tallying answers and counting the (sub) totals, where the researchers ascertained that the children were capable and enthusiastic about doing this. This survey led to a clearer and more detailed insight into the themes that were of importance to children living in neighborhood C. Because of this clearer insight, it was easier for the children to come up with feasible actions. The survey helped to prioritize and become more specific on what they wanted to change in the neighborhood. For example, the participating children believed that the price of soft drinks in the public ‘rainbow playground’ could be increased, to discourage children from buying sugary drinks. However, when asking their classmates, it appeared that a large share of the respondents did not think increasing the price would be a good idea. For that reason, there was no action undertaken on this idea and alternative ideas were thought out, e.g., asking the playground’s janitor to offer more healthy alternatives.

The setting in which the photovoice meetings took place influenced the proceeding of steps 3 and 4. As mentioned before, the setting in neighborhoods B and C was more school-like, in an enclosed space. Consequently, the focus group discussion meetings were structured, and because of the enclosed space, the children tended to listen to each other more since they were not easily distracted by other activities and people in the community center. The setting in neighborhood A was freer. This resulted in more chaotic proceedings, according to the researchers. These focus group discussion meetings required more attention from the children. Nonetheless, for the children of that specific neighborhood, this was the most expedient way, according to the community workers: *‘If you set too many rules, those children will stay away. Then they will be like “I come here because I like it”.’* (Community worker). In the researcher’s experience, a more voluntary setting also has the advantage that the children who actively choose to participate are really motivated. 

We present the themes and conclusions that were derived from the photovoice meetings and actions that followed as an illustrative example in [Appendix A]. Although showing those results was not the main aim of this paper and they are not generalizable to other contexts, they are interesting to give insight into the potential yields of such a project.

### 3.4. Step 5—Identifying Feasible Actions

The children all indicated that they enjoyed participating in project MAPZ because the project was action-oriented: *‘Like last week, with the surveys at school. We come up with something, and then we’re actually going to do it.’* (Child). Coming up with actions was not always easy, according to the community workers, because it was difficult for the children to see what lies within their sphere of influence. Children easily refer to adults when something should change in their opinion: ‘*They tend to point to others or organizations. They just don’t know, what can I influence as a child?’* (Community worker). In the researchers’ experience, the participating children needed help to devise actions that children could carry out themselves. During the meetings, researchers and community workers regularly asked the children: ‘But what can we do ourselves?’. When the group of children, researchers, and community workers were not able to carry out actions themselves, other parties (like the local government) were contacted. For example, the participating children concluded that the public ‘rainbow playground’ is too often closed. The action implemented to solve this issue was to write a letter to the playground janitor and have a conversation with him. The problem according to the playground janitor was that there were not enough volunteers to open the playground in more moments. The children offered to help recruit volunteers.

In the actions that were subsequently implemented, it is important, according to the community workers and researchers, that there would be a tangible result. If results would not be visible, children would find it harder to understand the rationale. Showing results also prevented children from suffering participation fatigue in the future: *‘Otherwise, when children are asked to participate the next time, they will think “well, nothing changed the last time, so, no*”.’ (Community worker). To prevent this from happening, it is important to formulate small and tangible actions. For example, one of the conclusions was that garbage and glass on playgrounds hindered children from being active outside. The children created (and hung up) warning signs right away.

Moreover, in the experience of the researchers, the participating children came up with creative and out-of-the-box solutions. For example, the children in neighborhood C concluded in the research that because of insufficient garbage bins in the neighborhood, a lot of garbage was on the streets. This made being physically active outside less attractive, according to the children. The local government removed garbage bins because they did not have enough capacity to empty them. The children came with a more creative action and set up the action ‘adopt a garbage bin’. They handed out flyers to people living in the neighborhood and asked them to adopt a garbage bin. Adopting a bin would entail emptying the bin when it is empty.

Coming up with actions themselves created a certain sense of ownership of the identified issues among the children. According to the community workers, they felt responsible for the problem and the solution since they contributed to an intervention themselves: *‘If they make something for their own environment, a sign against dog poo for example, and they can hang that up, then they feel proud. And then they will really make others aware of the need to throw dog poo away.’* (Community worker). According to the researchers and community workers, it was important that the children learned that they could influence their own environment: *‘You have made them realize, you can change the neighborhood. You are responsible for your neighborhood.’* (Community worker).

## 4. Discussion

In this PAR study, we showed how children aged 8–12 years living in three different low-SEP neighborhoods could participate in all phases of project MAPZ, and what, according to them and community workers and researchers, affected this process. Children participated throughout the whole process: from design to implementation and evaluation, in collaboration with the local welfare organization.

The finding that photovoice is an eligible method for the participation of children from low-SEP neighborhoods resonates with other photovoice studies with children [34,35]. The literature recognizes photovoice as an effective method for vulnerable youth to explore and describe their environment because children enjoy taking photos and showing them makes it easier to talk about their daily life [35,49]. In this study, photovoice made it possible to identify eligible opportunities for creating a healthier neighborhood environment. This was also found in other photovoice studies [50,51,52,53]. Opportunities for changing the environment found in this study can contribute to increased physical activity and healthy dietary behavior [15]. We showed that through PAR and photovoice, children were able to come up with out-of-the-box solutions. In addition, other studies in which target groups participated concluded that participatory research leads to more creative and novel solutions for complex issues [54,55]. Our finding on the importance of discussing the procedures of the project constantly with the community workers and children is in line with the conclusion in a qualitative review on photovoice [56]. Hergenrather et al. conclude that intensive collaboration between researchers and citizens and professionals is needed in order to match the project with the concerns and priorities of the participants [56]. An integrative review of PAR methodology and outcomes [39] shows that including children in the early stages of program development positively influenced the incorporation of their needs into the program, as a result of which developed programs are more appropriate for them.

Project MAPZ started in three different neighborhoods with a similar protocol, but the project proceeded differently. PAR projects are never a ‘one size fits all’ approach; they should always be adapted to the participants and local context [57]. Shamrova and Cummings [39] show that creating a methodology without addressing the abilities of the children to participate is difficult, especially in cases when the development of children’s participatory skills by the researchers is an inevitable part of PAR methodology. For example, in our study, one group consisted of children with a migrant background, who found it more difficult to express their perceptions in words compared with the other groups. Because less information was derived from this group of participants, a survey was conducted on the initiative of the children themselves among their classmates to verify the derived themes and gather more information on their views on their (un)healthy neighborhood. This shows that flexibility in the PAR process is needed.

The settings in which the PAR meetings took place influenced the research process. The importance of research settings when conducting research with children is discussed in previous research and shows that a formal school setting may be structured more hierarchically emphasizing adult-child imbalances of power [38]. A setting typically regulated by adults can lead to less creativity in children. In our project, this more school-like setting led to more structured meetings in neighborhoods B and C, but not to noticeably less creativity.

Children participating in the project learned a lot, e.g., to identify problems in the neighborhood and convert them into conclusions and actions. Researchers and community workers involved in this project noticed that by participating in the project, the children felt empowered to influence their own neighborhood environment. This is in line with the conclusion in a recent PAR study aimed at developing actions targeting healthy physical activity and dietary behavior among children [31]. Participation of children in their study led to critical awareness, leadership, and collaboration skills among participants, which contributed to increased feelings of empowerment. This is also in line with the vision of stimulating active citizenship through participation that is encouraged with children from an early age onwards in schools and communities [58].

### 4.1. Strengths and Limitations

A strength of this study is that we evaluated project MAPZ in three different low SEP neighborhoods. This made it possible to learn about the influence of the local context and characteristics of the participants on the PAR process. Results from local PAR studies are not easily generalized for other contexts [59]. By describing the local context, characteristics of the participants, and the procedures in the methods section, we hope that similar projects can translate our findings to their specific context and learn what important aspects should be given attention.

Participation of children in all phases of the project is another strength of this study. In participation projects, children are often only involved for the purpose of collecting data. Actively involving children in the design phase of the project leads to better alignment of the project to the children’s preferences, needs, and talents [56]. Through involving the children in interpreting the data and coming up with actions, the actions are more feasible, and the children feel more responsible for the solutions.

A limitation of this study is that due to the resignation of the community worker who facilitated the meetings in neighborhood B, the project ended abruptly in this neighborhood. As a result, the actions in neighborhood B were not implemented and no final evaluation with the children was performed. A disadvantage of collaboration with an organization is being dependent on the availability of their personnel and resources. To prevent this, it is good to properly safeguard and document the processes in the project and make sure that dependency on others involved in the project is minimized and that there is always a backup who can take over tasks. It is important that this newly involved professional takes time to build trust with participants.

### 4.2. Implications and Future Research

An implication of this study for the Zwolle Healthy City approach was that the organizations involved were enthusiastic about the photovoice method and were positive about using the method for engaging citizens in the development and implementation of their activities and policies in the future. The lessons learned were used as input for local policy on how to involve citizens in intervention and policy development.

For every new project, it is important to decide whether participation of the target group is possible and feasible. Moreover, methodological quality and scientific integrity should always be taken into account when deciding on the degree of participation [60]. In this study, the photovoice method turned out to be a suitable method for children. However, it took more time and effort to explain to the youngest children (those of 8 and 9 y/o) what they were asked to do in a photovoice task in comparison to the older children (those aged 10 and over).

Another implication for future PAR projects would be to involve more stakeholders in the community (e.g., policymakers and schools) from the start of the project. Even if it is unclear at the start of the project what the conclusions will be, and what stakeholders are needed in the action phase, it is always good to make contact with organizations in the community in an early stage of the project. Involving parents of the participating children could also be an implication for future PAR projects. Parents are genuinely interested in the activities of their children, and through the children, parents could be reached more easily. This offers opportunities to even involve parents in low-SEP neighborhoods, who are generally harder to reach.

The current PAR project ended with the implementation of actions and evaluation of the steps that lead to the actions. For future research, it would be interesting to evaluate the action phase and the consequences of the actions. Investigating participants’ levels of empowerment and how much the actions actually contributed to more healthy behavior, for example, is relevant.

## 5. Conclusions

In this PAR study, we showed how children aged 8–12 years living in three different low-SEP neighborhoods in the Netherlands could participate in all phases of project MAPZ, and what, according to them and community workers and researchers, affected this process. Children participated throughout the whole process: from design to implementation and evaluation, in collaboration with the local welfare organization. The photovoice method provided comprehensive information from the children’s perspectives. With the help of the community workers, the children identified feasible actions to improve the neighborhood environment. We found that it is important to constantly discuss the process with participants, start with a concrete question or problem, and adapt the project to the local context and skills of participants.

## Figures and Tables

**Table 1 ijerph-18-12131-t001:** Characteristics of participating children.

	Neighborhood A	Neighborhood B	Neighborhood C
	Youth club	Girls club	Children’s council
	*n* = 10	*n* = 11	*n* = 7
**Gender**			
Girl	5	11	6
Boy	5	0	1
**Age**			
8 years	4	0	1
9 years	2	2	0
10 years	2	2	1
11 years	1	3	1
12 years	1	4	4
**Nationality**			
Dutch	10	7	0
Migrant, non-western	0	4	7

**Table 2 ijerph-18-12131-t002:** Examples of questions asked in a focus group discussion with children.

Could you tell me about this photo?
Why did you take this photo?
Do you like or dislike what we see in the photo?
Would you like to change anything about what we see in the photo?
If so, what would you like to change about it?
Are you able to be physically active/play in this area?
Why (not)?
Are you able to make healthy food/drink choices here?
Why (not)?

**Table 3 ijerph-18-12131-t003:** Participants in the different types of data collection.

	Neighborhood A	Neighborhood B	Neighborhood C
	Youth club	Girls club	Children’s council
Step 1—Getting started	Community workers (*n* = 1) and researcher (*n* = 1)	Community worker (*n* = 1) and researcher (*n* = 1)	Community workers (*n* = 3)
Step 2—Taking photos	Children (*n* = 10)	Children (*n* = 11)	Children (*n* = 7)
Step 3—Identifying themes	Children (*n* = 10)	Children (*n* = 11)	Children (*n* = 7)
Step 4—Creating posters to draw conclusions	Children (*n* = 8)	Children (*n* = 10)	Children (*n* = 7)
Step 5—Identifying feasible actions	Children (*n* = 9)	Children (*n* = 10)	Children (*n* = 7)
Focus group discussion to evaluate	Children (*n* = 8)	-	Children (*n* = 7)
Semi-structured interview to evaluate	Community workers (*n* = 1) and researcher (*n* = 1)	Community worker (*n* = 1) and researcher (*n* = 1)	Community workers (*n* = 3)
Observations in all steps	Researcher (*n* = 2)	Researcher (*n* = 2)	Researcher (*n* = 1)
Informal chats after meetings in all steps	All	All	All

## Data Availability

All qualitative data files are available from the Data Archiving and Networking Services (DANS) repository database and can be found via https://doi.org/10.17026/dans-zh5-bhvr. Accessed on 16 November 2021.

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
