# Peer review of "Involving Children in Creating a Healthy Environment in Low Socioeconomic Position (SEP) Neighborhoods in The Netherlands: A Participatory Action Research (PAR) Project"

_ijerph, 2021, doi:10.3390/ijerph182212131_

Round 1

Reviewer 1 Report

Many thanks for this pleasant to read manuscript. It describes very well the method used (PAR) to include the target population in the design of an intervention. You lay out very well the process of interviewing the children from the different neighborhoods in Zwolle. As a previously purely quantitative researcher, I still managed to access the material very well. Regarding the implementation of the method, I was left with very few questions and see little need for revision of the manuscript in its current form, but I am not sure to what extent it is at all suitable for the IJERPH. In terms of content, it focuses exclusively on the methods used, and references to environment and health are minimal. That PAR and the Photovoice method are useful in developing an intervention to improve the health of children and adolescents in disadvantaged neighborhoods is indisputable, but the latter would also be entirely interchangeable in this manuscript. Health-related findings are ultimately only touched upon, and the method presented could be used in many other settings.

If I were to be stricter and evaluate the fit to the framework of the journal, I would probably have to say that this article would be better off in a journal focused on methods. Ultimately, however, that is left to the editors of IJERPH. I strongly recommend the following improvements:

Linguistically, there are some errors, especially "false friends" and similar mistakes (e.g., expressions that presumably can only occur when translated literally from Dutch into English, confusion of there/their). If the article were to be published in IJERPH, I would make the introduction and discussion stronger with health research in mind. That the method is good at reaching the target population of an intervention and incorporating their preferences in its development was well presented. Why would this be particularly helpful with regard to health interventions? What are other approaches to engaging the target population in health interventions and how are they better or worse? Can the health-related findings of the children's survey be better elaborated? How did the children's wishes differ from the experts' intentions? Also, was any of this implemented in the larger-scale intervention (i.e., beyond the smaller interventions implemented directly with the children)? 

Overall, I appreciated this manuscript very much.

Round 2

Reviewer 1 Report

Thank you very much for this revised manuscript.

I find the additions to the manuscript well done and now underline more clearly the relevance of the method in use for health research. This addresses what I consider to be the most important point regarding the "improvement" of the manuscript: the relevance for the journal. I also appreciate the linguistic improvements to the manuscript. Now, as a non-native English speaker, I can't really offer any more suggestions for improvement. I guess that - as it is almost always the case - there is still a small, but no longer significant room for improvement. In one case in line 356 you corrected a direct quote from an interview. I was wondering about that. Did you misquote the community worker before, or did you accidentally correct his/her expression?

Author Response

Dear reviewer 1, 

Thank you again for taking the time to read the revised manuscript. We're glad that we could succesully adress the points you raised before.

About the quote in line 356. Reviewer 2 pointed out that the verbatim quote in the first version of the manuscript was incorrect English language. The Dutch quote of the community worker was translated wrong. The correction we made was not accidently, but was made to improve the English language. 

We hope this is clear now.

Sincerely, 

Lisa Wilderink and colleagues